# Facile Generation of Potent Bispecific Fab via Sortase A and Click Chemistry for Cancer Immunotherapy

**DOI:** 10.3390/cancers13184540

**Published:** 2021-09-10

**Authors:** Xuefei Bai, Wenhui Liu, Shijie Jin, Wenbin Zhao, Yingchun Xu, Zhan Zhou, Shuqing Chen, Liqiang Pan

**Affiliations:** 1Institute of Drug Metabolism and Pharmaceutical Analysis, College of Pharmaceutical Sciences, Zhejiang University, Hangzhou 310058, China; xfbai@zju.edu.cn (X.B.); liuwenhui@bangshunpharm.com (W.L.); jinsj08@163.com (S.J.); pharmacy_zwb@zju.edu.cn (W.Z.); ycxu66@163.com (Y.X.); zhou@zju.edu.cn (Z.Z.); 2Hangzhou Biosun Pharmaceutical Co., Ltd., Liangzhu International Life Science Town, 268 Tongyun Street, Yuhang District, Hangzhou 310015, China; 3Collaborative Innovation Center of Artificial Intelligence by MOE and Zhejiang Provincial Govement (ZJU), Hangzhou 310058, China; 4Department of Precision Medicine on Tumor Therapeutics, ZJU-Hangzhou Global Scientific and Technological Innovation Center, Hangzhou 311200, China; 5Key Laboratory of Pancreatic Disease of Zhejiang Province, The First Affiliated Hospital, School of Medicine, Zhejiang University, Hangzhou 310003, China

**Keywords:** bispecific antibody, sortase A, chemo-enzymatic approach, anti-CD20 antibody, Fab, BiFab

## Abstract

**Simple Summary:**

The formats of bispecific antibody have been investigated for many years to enhance the stability of the structure and anti-tumor efficacy. One of the formats combining two Fabs at their C termini provides unmodified variable region and comparable activity to other fragment-based bispecific antibodies that are usually combined in a head-to-tail manner. However, the current strategy to produce the BiFab molecule is limited to a semisynthetic method that introduces unnatural amino acid to antibodies’ sequences during production. To improve the application of BiFab format in investigational biodrugs, we have applied sortase A-mediated “bio-click” chemistry to generate BiFab, for facile assembly of Fab molecules that have been expressed and stored as BiFab module candidates. The BiFabs made by our method stimulate T cell proliferation and activation with favorable in vitro and in vivo anti-tumor activit. Our results indicate that BiFab made by sortase A-mediated click chemistry could be used to efficiently generate various BiFabs with high potency, which further supports personalized tumor immunotherapy in the future.

**Abstract:**

Bispecific antibodies (BsAbs) for T cell engagement have shown great promise in cancer immunotherapy, and their clinical applications have been proven in treating hematological malignance. Bispecific antibody binding fragment (BiFab) represents a promising platform for generating non-Fc bispecific antibodies. However, the generation of BiFab is still challenging, especially by means of chemical conjugation. More conjugation strategies, e.g., enzymatic conjugation and modular BiFab preparation, are needed to improve the robustness and flexibility of BiFab preparation. We successfully used chemo-enzymatic conjugation approach to generate bispecific antibody (i.e., BiFab) with Fabs from full-length antibodies. Paired click handles (e.g., N_3_ and DBCO) was introduced to the C-terminal LPETG tag of Fabs via sortase A mediated transpeptidation, followed by site-specific conjugation between two click handle-modified Fabs for BiFab generation. Both BiFab^CD20/CD3^ (EC_50_ = 0.26 ng/mL) and BiFab^Her2/CD3^ exhibited superior efficacy in mediating T cells, from either PBMC or ATC, to kill target tumor cell lines while spared antigen-negative tumor cells in vitro. The BiFab^CD20/CD3^ also efficiently inhibited CD20-positive tumor growth in mouse xenograft model. We have established a facile sortase A-mediated click handle installation to generate homogeneous and functional BiFabs. The exemplary BiFabs against different targets showed superior efficacy in redirecting and activating T cells to specifically kill target tumor cells, demonstrating the robustness of sortase A-mediated “bio-click” chemistry in generating various potent BiFabs. This approach also holds promise for further efficient construction of a Fab derivative library for personalized tumor immunotherapy in the future.

## 1. Introduction

Immunotherapies, such as chimeric antigen receptor T cells (CAR-Ts) and T-cell-engaging bispecific antibodies (T-BsAbs), have revolutionized cancer treatments by leveraging the immune system [1,2]. T-BsAbs usually refer to bifunctional antibodies with one arm targeting T cell receptors (e.g., CD3) to engage T cells and another arm targeting antigen on tumor cells, for the purpose of bridging and redirecting T cells to tumor cells. Compared with CAR-T cells, which are autologous T lymphocytes that are genetically engineered to express chimeric antigen receptor for specific tumor cell targeting [3], bispecific antibody can be produced relatively easier and provide off-the-shelf treatment [4,5]. This strategy has generated great interest with more than 50 T-BsAb candidates in clinical trials for a range of indicators nowadays [6].

One representative T-BsAb is Blinatumomab, a bi-specific T cell engager (BiTE) targeting CD19 and CD3 that was approved by FDA in 2014 for the treatment of Acute Lymphoblastic Leukemia (ALL). The flexible tandem arrangement of this single chain bispecific antibody accounts for its superior efficacy in inducing lytic synapse and thereby high T cell activity in comparison with its IgG-based and Fab-based format [7,8]. Despite the high efficacy, BiTE molecule has a very short half-life of ~2 h in blood circulation in the absence of Fc domain [9]. In order to increase stability and activity of fragment-based T-BsAbs, Dual-Affinity Re-Targeting (DART^®^) protein and tandem diabody (TandAb) were designed to further improve the half-life and stability in vivo [10,11,12,13]. However, the variable region spanning engineered constant scaffold might result in the loss of affinity and stability for Fc-free T-BsAbs, such as single-chain variable fragment (scfv) molecules [14,15]. For example, variable regions assembled to a format that deviate significantly from its cognate high stable IgG might compromise its affinity, especially when the N-terminus of Fvs have additional polypeptide chains that function as linkers [14,16,17].

Bispecific antibody binding fragment (BiFab) represents another promising platform for generating bispecific antibodies. Two Fab fragments providing different binding specificities are usually chemically linked in a tail-to-tail manner to generate BiFab. The intact structure of Fab fragments is parallelly grafted into the BiFab format, which maintains a natural association of four domains (VL, CL, VH and CH1) and thus ensures stability [14,18,19]. The BiFabs could also avoid Fc-related side effects since they lack a Fc region. However, the site-specific conjugation of two Fab molecules remains challenging during BiFab preparation [20]. One of the well-known chemical approaches for BiFab generation is the application of click chemistry, in which the click handle is installed through the introduced noncanonical amino acid (ncAA) on Fabs, to realize site-specific conjugation of two Fabs [21,22,23].

To achieve site-specific click handle installation, an alternative approach is sortase A-mediated transpeptidation. Sortase A is a bacterial enzyme that recognizes C-terminal LPXTG motif (X represents any amino acid) of proteins or peptides, which is used to anchor building blocks of cell walls of Gram-positive bacteria. The enzyme cleavages between Thr and Gly residues and then yields an acyl-enzyme intermediate. Subsequently, the nucleophilic primary amine of oligo-glycine modified substrates resolved the intermediate and then form a covalent bond between oligo-glycine modified substrates and LPETG-tagged protein [24,25,26,27]. Therefore, the paired click handles could be modified with oligo-glycines, such as GGG, before installation to the C-terminus of the target protein (e.g., Fab). Herein, we applied sortase A-mediated two-step chemo-enzymatic conjugation to generate BiFabs. The paired click handles that comprising azide and dibenzocyclooctyne function groups was firstly attached to the Fabs by sortase A mediated transpeptidation between LPETG-tagged Fab and click chemistry-functionalized GGG, and subsequently the Fab-linkers are conjugated via click chemistry to form BiFabs. Using this strategy, we successfully constructed homologous BiFab^CD20/CD3^ and BiFab^Her2/CD3^. We have demonstrated the potent in vitro and in vivo efficacy of BiFab^CD20/CD3^, and its ability to stimulate resting PBMC to proliferate and degranulate. In addition, functional BiFab^Her2/CD3^ was generated by simply replacing Fab^CD20^ arm with Fab^Her2^, further suggesting the potential of this chemo-enzymatic approach on preparing various BiFabs based on prestored Fab derivative library.

## 2. Materials and Methods

### 2.1. Reagents and Cell Lines

The human CD20-positive cell lines Ramos, Raji, Daudi and the human CD20-negative cell line K562 were purchased from the American Type Culture Collection (ATCC, San Francisco, CA, USA), and were cultured in 1640 medium (Gibco) with 10% fetal bovine serum (FBS, Gibco). The human HER2-positive cell line SK-OV-3 and HER2-negtive cell line MDA-MB-468 were purchased from ATCC and were cultured in McCoy’s 5A or DMEM (Gibco) with 10% FBS, respectively. The expression plasmids of the full-length anti-CD20 antibody Ofatumumab and sortase A enzyme were constructed in our laboratory [28]. The HEK-293F cell line was from Qilin Zhang’s laboratory in Tsinghua University. The HEK293F cells were grown in 250 mL SMM-293-TI medium (Sinobiological, Beijing, China) supplemented with 100 U/mL ampicillin, 100 μg/mL streptomycin (Sorlabio), and 1% FBS and the cells were shaking cultured at 37 °C and 210 rpm (Eppendorf). Anti-CD3 Fab sequence was derived from the humanized OKT3 antibody [29]. Anti-Her2 Fab sequence was derived from the Trastuzumab [30].

Triple glycine-modified linker Gly_3_-(PEG)_3_-N_3_ (GPN) were synthesized by Concortis (San Diego, CA, USA). Triple glycine-modified linker Gly_3_-(PEG)_4_-dibenzocyclooctyne (DBCO) (GPD) was purchased from Lumiprobe (Hunt Valley, MD, USA).

### 2.2. Sortase A-Mediated Click Handle Installation

We previously showed that sortase A was used to specifically conjugate LPETG tagged IgG with GGG modified toxins [24], and the enzyme was kept by our lab. Briefly, we used a sortase A mutant (△N59) derived from *Staphylococcus aureus*, which is subcloned into pET28a(+) before a six Histidine polypeptide (His_6_). The expression vector of sortase A was then transfected into BL21 (DE3) Competent Cells (Sangon, Shanghai, China) and the expression is induced by 0.5 M IPTG for 16 h. After incubation, cells were harvested and disrupted by French Press (ThermoFisher Scientific Inc., Shanghai, China). The soluble fraction was collected and purified by Ni-NTA (HiTrap Ni-NTA column, GE) with instruction of the manufacturer’s protocol. The purified sortase A protein was buffer exchanged to 50 × 10^−3^ M Tris–HCl (pH 7.5), 150 × 10^−3^ M NaCl by ultrafiltration (Amicon Ultra-10k, Millipore, MA, USA), sterile filtered and stored at −80 °C. Sequences of light chain and heavy chain of antibody fragments (Fabs) were, respectively, inserted into pMH3 expression vector behind human signal peptide sequence, and Fabs of heavy chain were C-terminally tagged with nucleotide sequence that express polypeptide GGGGSGGGGSGGGGS-LPETG-6 × His ((G_4_S)_3_-LPETG-His_6_). G_4_S linker was used to facilitate sortase A mediated transpeptidation. The expression vector of Fabs was transiently expressed in HEK293F cells for 3–4 days.

To optimize the reaction conditions of the sortase A-mediated conjugation, the reaction molar ratio of antibody fragments to glycine modified linkers (e.g., GPD and GPN) was explored. The reaction molar ratios (1:25 and 1:50) and different reaction time (6 h, 12 h or 24 h) at 37 °C were investigated in reaction buffer (50 mM Tris-HCl, 150 mM NaCl, 5 mM CaCl_2_, pH 7.4) solution in the presence of 50 μM sortase A enzyme (the molar ratio of sortase A/Fab was 1:8.3). To evaluate the conjugation efficiency, the reverse-phase high pressure liquid chromatography (RP-HPLC) with a Varian PLRP-S 100 Ǻ column was used as previously described [28,31]. The conjugation reaction was scaled up under optimal reaction condition. Since the His tag was cut off by sortase A during transpeptidation, the flow-through fluid containing modified Fabs (e.g., Fab^CD3^-DBCO and Fab^CD20^-N_3_) was collected during HiTrap Ni-NTA affinity chromatography. All modified Fabs were buffer exchanged to PBS (pH 7.4) by ultracentrifugation (Millipore Amicon Ultra Filters, 10 kDa cut-off).

### 2.3. Click Chemistry Mediated Generation of Bispecific Fab (BiFab)

The copper-free click reaction between Fab-GPN and Fab-GPD was reacted in a buffered solution contained 50 mM Tris-HCl, 150 mM NaCl (pH 7.4). Fab^CD3^-DBCO was reacted with Fab^CD20^-N_3_ or Fab^Her2^-N_3_ at a molar ratio of 1:1 at 4 °C for 12 h. After reaction, BiFabs were purified from free Fab by size exclusion chromatography (SEC) (Superdex 200 increase 10/300 GL, GE) on AKTA purifier (Amersham Biosciences, MA, USA). Sample from each peak was analyzed by SDS-PAGE under reducing condition and non-reducing condition. The purified protein from SEC was also analyzed by RP- HPLC with the following condition, a linear gradient elution starting from 75% buffer A (1.5 M (NH_4_)_2_SO_4_, 25 mM Na_3_PO_4_, pH 7.0), 25% buffer B (25 mM Na_3_PO_4_, pH 7.0) and 0% isopropanol, to 0% buffer A (1.5 M (NH_4_)_2_SO_4_, 75 mM Na_3_PO_4_, pH 7.0), 75% buffer B (25 mM Na_3_PO_4_, pH 7.0) and 25% isopropanol.

### 2.4. Flow Cytometry

All flow cytometry studies were conducted on ACEA NovoCyteTM (ACEA Biosciences Inc., San Diego, CA, USA). Data were processed with FlowJo 10.1 (FlowJo, LLC, Ashland, OR, USA) and Prism 8.0.1 (GraphPad Software Inc., San Diego, CA, USA).

To evaluate the binding ability of BiFab^CD20/CD3^, 1 × 10^6^ CD20-positive cells or 1 × 10^6^ CD3-positive Jurkat cells were incubated with serial concentrations of Fab^CD20^, Fab^CD3^ and BiFab^CD20/CD3^ in ice-cold PBS (pH 7.4) for 30 min, followed by incubation with the primary anti-human IgG-Fab fragment (Abcam, Cambridge, UK) for 30 min. After washing three times with cold PBS (pH 7.4), cells were incubated with secondary goat anti-mouse IgG-FITC (Beyotime, Shanghai, China) for 30 min. After washing step, immune-stained cells were analyzed by flow cytometry.

### 2.5. Preparation of Active T Cells (ATC) from Peripheral Blood Mononuclear Cells (PBMC)

Human blood samples were obtained from healthy volunteers. PBMC were extracted from fresh blood samples by density centrifugation (Ficoll-Paque) following manufacturer’s instruction.

PBMC were stimulated with Dynabeads™ Human T-Activator CD3/CD28 (Thermo Fisher) for T cell expansion and activation to generate active T cells (ATC). Briefly, PBMC were mixed with dynabeads at a cell-to-bead ratio of 1:1, and co-incubated for 4 days in the presence of 30 U/mL recombinant IL-2.

### 2.6. Cell Apoptosis

PBMCs were used as effector cells in all experiments. For LDH releasing assay, 96-well plates were seeded with 3 × 10^4^ tumor cells (e.g., Ramos or Daudi cells) and 6 × 10^4^ ATC per well, and then added with serial concentrations of BiFabs for a 24 h incubation at 37 °C. After incubation, the release of the intracellular enzyme lactate dehydrogenase (LDH) was determined by LDH cytotoxicity assay kit (Beyotime, Shanghai, China) to measure cell death. The percentage of necrotic cells was calculated according to the absorbance of each well at 450 nm.

For flow cytometry studies on cell apoptosis, ATC were prestained with Carboxyfluorescein succinimidyl ester (CFSE), and then co-cultured with 2 × 10^5^ tumor cells at an effector: target (E:T) ratio of 2:1 for 24 h. When PBMC were used as effector cells, the E:T ratio was 5:1. Cells were then stained with Annexin-Cy5/Propidium Iodide (PI), and the percentage of apoptotic and necrotic cells were determined by flow cytometry.

### 2.7. T Cell Activation

CD69 and CD25 are early and late activation markers for T cells, respectively. We therefore used flow cytometry to evaluate T cell activation via measuring cell surface CD69 and CD25 expression. Fresh PBMC were mixed with target tumor cells (e.g., Ramous, Raji, Daudi and K562 cells) at E: T ratio of 5:1 before adding serial concentrations of BiFabs (BiFab^Her2/CD3^ or BiFab^CD20/CD3^) to initiate specific killing, and the co-incubation lasted 48 h. Naïve T cells were labeled with FITC-αCD4 and FITC-αCD8, and active T cells were further labelled with APC-αCD25 and PE-αCD69 (BD Biosciences). When CD20 positive tumor cells were used as target cells, fresh PBMC were pre-treated with anti-CD20 antibody-coated magnetic beads to deplete CD20-positive B cells.

Enzyme-linked immunosorbent assay (ELISA) was used to detect interferon gamma (IFN-γ) that was secreted from the activated T cells. Fresh PBMC were co-incubated with target tumor cells (e.g., K562 cells) at an E: T ratio of 5:1, and then treated with BiFabs or IgG format bispecific antibodies for 48 h. Supernatants were collected for IFNγ detection through Human IFN-γ CytoSetTM KIT (Invitrogen, Shanghai, China). The absorbance at 450 nm was measured by 680 Microplate reader (Bio-Rad, Hercules, CA, USA).

To evaluate T cell proliferation after activation, fresh PBMC were pre-labelled with CFSE and then mixed with target tumor cells at an E: T ratio of 5:1. The cell mixtures were treated with different concentrations of BiFabs for 48 h. T cell proliferation was further determined by flow cytometry. Ramos cells were used in experimental groups as CD20-positive cells, while K562 cells served as CD20-negative cell control.

### 2.8. In Vivo Antitumor Activity of BiFab in Mouse Xenograft Model

Eight-week-old female SCID Beige mice were inoculated subcutaneously with 2.5 × 10^6^ Ramos cells and 1 × 10^7^ PBMC (E:T = 4:1) into the right flank of the nude mice. Inoculated mice were randomly divided into 4 groups: vehicle group, Fab^CD3^ group, Fab^CD20/CD3^ (3 mg/kg) group and BiFab^CD20/CD3^ (1 mg/kg) group. Mice in the experimental groups received BiFab^CD20/CD3^ (1 mg/kg or 3 mg/kg) by intravenous (i.v.) injection into tail vein. Mice in the control groups received Fab^CD3^ (1 mg/kg) or saline. Each treatment was given four times at 2-day intervals (q2d × 4). The mean tumor volume and mouse body weight were measured using calipers and an electronic balance, respectively. The mean tumor volume was calculated using the formula: tumor volume (mm^3^) = tumor length × tumor width × tumor width/2.

### 2.9. Statistical Analysis

Statistical analysis was performed by using GraphPad Prism 6.01 software. Student’s *t*-test was used when two independent groups are compared, while Dunnett’s multiple comparison test was used for comparison of multiple groups. Statistical significance was determined by the *p* value (* *p* < 0.05, ** *p* < 0.01, and *** *p* < 0.001).

## 3. Results

### 3.1. Generation of Bispecific Fab via Sortase-Mediated Transpeptidation and Click Chemistry

The whole procedure to generate BiFabs was summarized in Figure 1a. Fabs targeting CD20, CD3 or HER2 were first expressed with LPETG-His_6_ tail at C terminus of heavy chains (Figure 1b) and stored for future assembly after purification. GGG-PEG_3_-N_3_ or GGG-PEG_4_-DBCO was linked onto Fabs via sortase A transpeptidation, and His-tag was released from Fabs, which spared linker-Fab components from the reaction mixture when purified by Ni-NTA affinity chromatography. Before click reaction, the optimal molar ratio and reaction time for sortase A-catalyzed reaction was investigated. According to peak shifting of H-DBCO, the optimal reaction condition is 1:25 of Fab^CD3^ and GPD and reacted for 12 h (Figure 1c), in which there is much less unconjugated heavy chain (peak “H”) compared to other reaction conditions. Click reaction between Fab^CD3^-DBCO and Fab^CD20^-N_3_ at a molar ratio of 1:1 efficiently generated BiFab^CD20/CD3^. After click reaction, homogenous BiFab^CD20/CD3^ was obtained by size exclusion chromatography purification and further confirmed by SDS-PAGE (Appendix A). The assembly of Fab^Her2^ and Fab^CD3^ was conducted in the same way to generate homologous BiFab^Her2/CD3^ (Figure 1d). The purity of BiFab^CD20/CD3^ was further confirmed by RP-HPLC analysis (Figure 1e). According to the peak area, the content of BiFab^CD20/CD3^ in the final buffered solution is about 95% after SEC purification and ultraconcentration.

### 3.2. The Binding Ability of BiFabs with Target and Effector Cells

To confirm whether BiFab^CD20/CD3^ maintained the binding ability of two Fabs, we used Jurkat cells (CD3 positive) and Ramos cells (CD20 positive) for flow cytometric analysis of BiFab^CD20/CD3^. The BiFab^CD20/CD3^ showed concentration-dependent binding with CD20-positive Ramos cells and CD3-positive Jurkat cells (Figure 2a). Interestingly, BiFab^CD20/CD3^ had a higher binding affinity compared to that of Fab^CD20^ or Fab^CD3^ monomers to target cells (Figure 2a). Upon binding with CD3 on T cells and CD20 on tumor cells, the BiFab^CD20/CD3^ could efficiently activate T cells according to the measurement of cell-surfaceCD69 and CD25, which represent early and late activation markers on T cells, respectively (Figure 2b). At the same concentrations of BiFab^CD20/CD3^, the expression level of CD69 was much higher than that of CD25, which exhibited a quicker response curve of CD69 comparing to CD25. Similarly, the BiFab^Her2/CD3^ generated by replacing Fab^CD20^ could also bind to HER2-positive SK-OV-3 cells and CD3-positive Jurkat cells (Figure 2c).

### 3.3. BiFab Efficiently and Specifically Induced Cytokine Release and Proliferation of T Cells

The release of interferon-γ (INF-γ) was evaluated as this cytokine is essential for mediating the antitumor activity. We measured INF-γ release by ELISA kit with CD20+ Daudi and Raji cells as target cells and unstimulated PBMC as effector cells. In both types of target cells, high level of IFN-γ release was detected in the culture supernatants in the presence of BiFab^CD20/CD3^ (400 and 2000 ng/mL) (Figure 2d,e). We also noticed that BiFab^CD20/CD3^ induced stronger T cell activation at a concentration of 80 ng/mL when the target cells were Daudi cells in comparison to Raji cells. Almost no T cell activation was observed in the absence of BiFab^CD20/CD3^, suggesting the specific mode of action underlying BiFab mediated T cell engaging.

For the analysis of T cell proliferation after BiFab stimulation, fresh PBMC were pre-stained with CFSE, a cell permeant green fluorescent molecule whose succinimidyl ester group reacts indiscriminately and covalently with primary amines of intracellular proteins, to facilitate fluorescent labeling of T cell population. Upon incubating with BiFab^CD20/CD3^, PBMC significantly proliferated after 48 h in the presence of target cells (i.e., CD20-positive Ramos Cells) (Figure 2e). No obvious T cell proliferation was observed in negative control group (CD20-negative K562 cells). We further studied the proliferation rate with various concentrations of BiFab^CD20/CD3^ measured by flow cytometry. BiFab^CD20/CD3^ triggered T cells proliferation in a concentration-dependent manner (Figure 2f).

### 3.4. BiFabs Redirected T Cells to Kill Target Tumor Cells

In vitro cytotoxicity of BiFab^CD20/CD3^ was measured by LDH releasing assay on Daudi and Ramos cell lines. BiFab^CD20/CD3^ efficiently induced tumor cell apoptosis at an E:T ratio of 2:1, achieving half maximal-apoptosis rate at a concentration of 0.262 ng/mL (2.62 pM) on Daudi cells and 0.275 ng/mL (2.75 pM) on Ramos cells (Figure 3a). The apoptosis-inducing efficacy of BiFab^CD20/CD3^ was further assessed by FITC-Annexin V/PI staining assay on Daudi and Ramos cell lines. BiFab^CD20/CD3^ could induce maximal apoptosis, including early (Annexin V+/PI−) and late (Annexin V+/PI+) apoptotic cells, on Daudi cells at various concentrations (10 ng/mL–10 ug/mL) (Figure 3b). For Ramos cells, the apoptosis rate, ranging from 70–100%, was concentration-dependent and lower than that of Daudi cells (Figure 3b). Fc-mediated nonspecific activation through binding to Fc receptors on immune cells could probably cause toxicity [32]. Comparing to IgG^CD20/CD3^ which could elicit Fc-mediated non-specific killing, the BiFab^CD20/CD3^ was demonstrated to have minimal killing towards CD20 negative K562 cells, suggesting the advantage of Fc truncation in eliminating Fc-mediated side effects (Figure 3c). Similar to BiFab^CD20/CD3^, the BiFab^Her2/CD3^ exhibited remarkable killing efficacy on HER2-positive SK-OV-3 cells while spared HER2 negative MDA-MB-468 cells with marginal cell killing (Figure 3d). The results showed here suggested that sortase A-mediated chemo-enzymatic approach was successfully applied to the generation of other BiFabs.

### 3.5. BiFab^CD20/CD3^ Eliminated B-Cell Lymphoma in Xenograft Mouse Model

We next evaluated the in vivo efficacy of BiFab^CD20/CD3^ with mouse xenograft model of B-cell lymphoma. Mouse xenograft tumor model was successfully established by co-injection of Ramos and PBMC cells (E:T ratio = 4:1). The intravenous administration of BiFabs was initiated 24 h after inoculation to facilitate T cell activation. The administration was repeated every two days for a total of four injections. Strikingly, the BiFab^CD20/CD3^ completely suppressed the tumor growth at a dosage of 3 mg/kg, and there was only one mouse that underwent a recurrence in the 1 mg/kg group (Figure 3e). In contrast, the anti-CD3 Fab group did not show any significant efficacy in vehicle group, in which tumor grew rapidly. These results demonstrated that the BiFab^CD20/CD3^ could efficiently mediate T cell killing in vivo.

## 4. Discussion

We have presented here a facile approach utilizing sortase A-mediated bio-click chemistry to generate BiFabs with potent antitumor activity. Paired click handles (e.g., N_3_ and DBCO) was conjugated to the C-terminal LPETG tag of Fabs via sortase A mediated transpeptidation, followed by site-specific conjugation between two click handles-modified Fabs for BiFab generation. We have presented exemplary BiFabs against two different targets. First, the BiFab^CD20/CD3^ exhibited superior efficacy in mediating T cells, from either PBMC or ATC, to kill multiple CD20-positive lymphoma cell lines while spared CD20-negative tumor cells in vitro (Appendix A). The BiFab^CD20/CD3^ also efficiently inhibited CD20-positive tumor growth in the mouse xenograft model (Figure 3e). Second, the BiFab^Her2/CD3^ also showed potent in vitro antitumor activity against HER2-positive tumor cell lines (Figure 3d), demonstrating the robustness of sortase A-mediated bio-click chemistry in generating various potent BiFabs.

The first BiFab construct, termed as BsF(ab’)2, was first generated by chemical conjugation of Fab’-SH with the thionitrobenzoate derivative of another Fab (Fab’-TNB), which was described by Paul Carter et al. from Genentech Inc. [19]. The BiFab^CD20/CD3^ was also generated without Fc region. Fc region of IgG-based bispecific antibodies could potentially induce nonspecific T cell activation [33], causing off-target T cell engaging-related side effects. However, cytokine-related adverse effects, such as cytokine release syndrome (CRS), are probably inevitable for T cell engaging and activation related immunotherapy, e.g., CAR-T, BiTE [34,35]. At present, T-cell engagers targeting CD20 are mostly based on classical IgG-based antibody. Sun et al. [36] reported a T-cell recruiting bispecific antibody CD20-TDB with EC_50_ of 0.22–11 ng/mL at the E: T ratio of 10:1. Smith et al. [37] reported another anti-CD20/CD3 T cell engagers REGE2280 and REGN 1979, which showed favorable EC_50_ of 2.25–12.6 ng/L at the E:T ratio of 10:1 (ATC as effector cells). FBTA05 is a trifunctional chimeric rat/mouse CD3 × CD20 targeting bispecific antibody, and therefore it has higher immunogenicity [38]. In comparison with above anti-CD20/CD3 bispecific antibodies, our BiFab^CD20/CD3^ showed a potent apoptosis-inducing ability at an E:T ratio of 2:1 using ATC as effector cells (EC_50_ = 0.26 ng/mL for Daudi cell lines and 0.275 ng/mL for Ramos) (Figure 3a). The in vivo antitumor efficacy of BiFab^CD20/CD3^ was consistent with its in vitro efficacy, since four intravenous injections (3 mg/kg) of BiFab^CD20/CD3^ completely suppressed tumor growth in the mouse tumor xenograft models (Figure 3e).

We previously reported a nucleic acid (i.e., left-handed DNA, L-DNA) mediated protein-protein assembly (NAPPA) approach to offer a general approach for preparing antibodies with higher-order specificity [39]. Similar to the NAPPA approach, our two-step conjugation strategy allows the preparation of modular Fab derivatives and the generation of customized Fab library thereof, which is the major difference comparing with the conventional BiFab construction methods (Figure 4). In addition, both BiFab^CD20/CD3^ and BiFab^Her2/CD3^ showed potent antitumor efficacies, regardless of different tumor target, suggesting the effectiveness and robustness of sortase A mediated chemo-enzymatic approach. Lawrence G Lum et al. [40] explored the application of anti-CD20/CD3 bispecific anti-body-armed activated T cells (aATC). The anti-CD20/CD3 aATC was a cell-based therapy that activated T cells from patients were armed with chemically conjugated anti-CD3 × anti-CD20 bispecific antibody, and then expanded and re-infused into patients. The aATC therapy was demonstrated to be safe and effective in a phase I clinical trial [40]. Inspired by this study, the sortase A-mediated bio-click chemistry could be further applied to personalized immunotherapy through ATC armed with combination-optimized BiFab. Since the efficacy of BiFab varies when using Fabs with different affinities or paratopes, the sortase A mediated transpeptidation reaction during BiFab generation facilitates the construction of Fab library for rapid efficacy evaluation of different BiFabs.

## 5. Conclusions

We constructed BiFab^CD20/CD3^ and BiFab^Her2/CD33^ via sortase A-mediated bio-click chemistry and demonstrated their anti-tumor activity through engaging human immune cells. Our results shown here indicates that Sortase A-mediated click handle installation holds promise for facile generation of potent bispecific Fabs and further efficient construction of Fab derivative library for personalized tumor immunotherapy in the future.

## Figures and Tables

**Figure 1 cancers-13-04540-f001:**
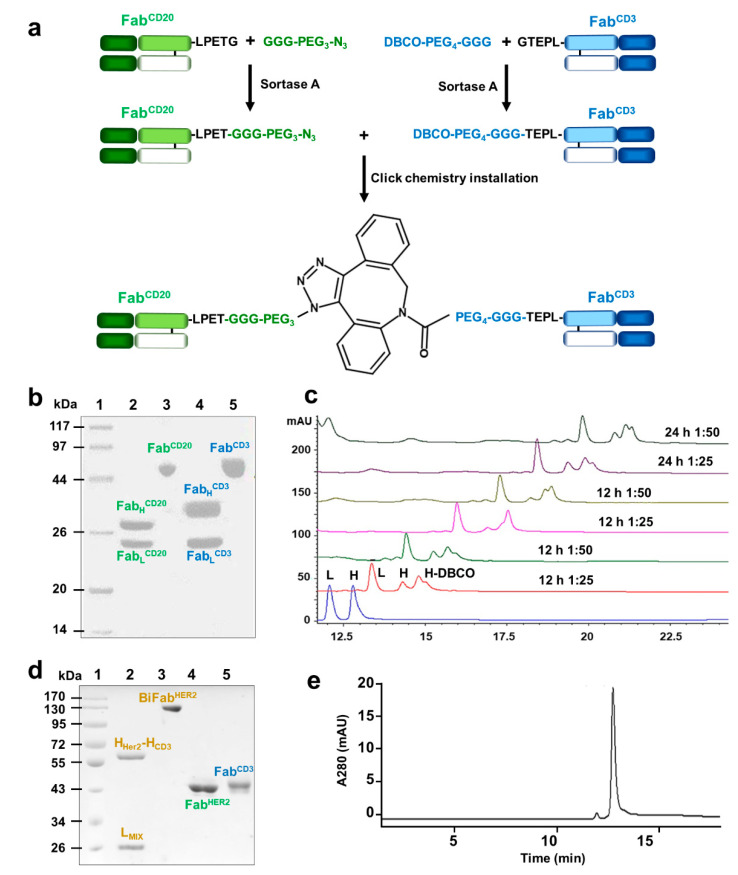
Generation and characterization of BiFabs. (**a**) Schematic diagram of sortase A-mediated click chemistry installation for BiFab preparation. (**b**) Characterization of the purified Fabs by SDS-PAGE. Lane 1, high molecular weight protein marker; Lane 2, the reduced Fab^CD20^; Lane 3, the intact Fab^CD20^; Lane 3, the reduced Fab^CD3^; Lane 4, the intact Fab^CD3^. (**c**) Reverse-phase HPLC analysis of Fab-click handle conjugation through sortase A-mediated transpeptidation, under different reaction conditions. (**d**) Characterization of BiFabs by SDS-PAGE. Lane 1, high molecular weight protein marker; Lane 2, the reduced BiFab^Her2/CD3^; Lane 3, the intact BiFab^Her2/CD3^; Lane 4, the intact Fab^Her2^; Lane 5, the intact Fab^CD3^. (**e**) Reverse phase high-performance liquid chromatography (RP-HPLC) analysis of the purity of BiFab^CD20/CD3^.

**Figure 2 cancers-13-04540-f002:**
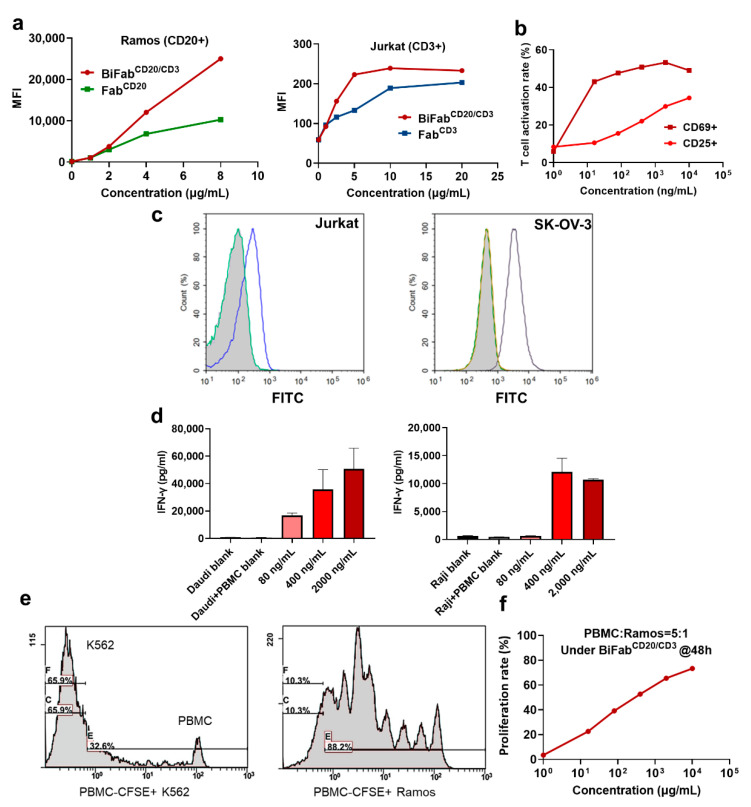
In vitro efficacy of BiFabs. (**a**) The binding abilities of Fabs and BiFab with CD20-positive Ramos and Jurkat cells. (**b**) The in vitro efficacy of the BiFab^CD20/CD3^ on T cell activation. After CD20-positive B cell depletion, fresh PBMCs were treated with serial concentrations of BiFab^CD20/CD3^ in the presence of target tumor cells at an E:T ratio of 5:1 for 48 h. The expression levels of CD69 and CD25 on T cells, two biomarkers for T cell activation, were evaluated after immuno-staining via flow cytometry. (**c**) Evaluation of the binding abilities of BiFab^Her2/CD3^ with CD3-positive Jurkat cells and HER2-positive SK-OV-3 cells by flow cytometry. (**d**) The quantification of interferon-γ release from T cells activated by BiFab^CD20/CD3^. Fresh PBMCs were incubated with Daudi or Raji cells at an E:T ratio for 5:1 for 48 h. The secreted interferon-γ from T cells was quantified by ELISA Kit. (**e**) BiFab^CD20/CD3^ mediated T cell proliferation in the presence of CD20-negative K562 cells or CD20-positive Ramos cells at an E:T ratio of 5:1 for 48 h. (**f**) After treatment with various concentrations of BiFab^CD20/CD3^ with an E:T ratio of 5:1 for 48 h, T cell proliferation was analyzed by flow cytometry.

**Figure 3 cancers-13-04540-f003:**
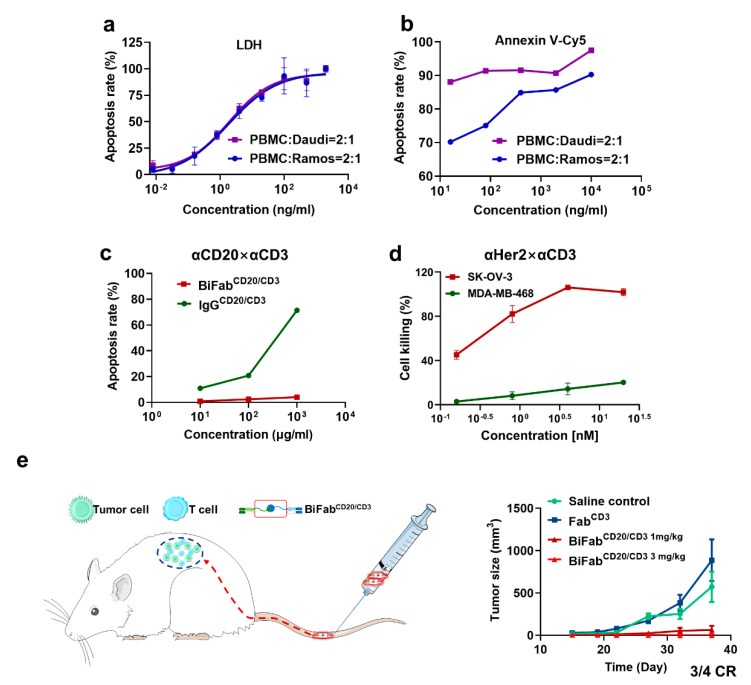
The in vitro and in vivo antitumor activities of BiFabs. (**a**) The in vitro efficacy of BiFab^CD20/CD3^. Target cells (Ramos and Daudi) and active T cells (E:T = 2:1) were incubated with serial diluted BiFab^CD20/CD3^ for 24 h (data shown as mean ± SD, *n* = 3). LDH release was determined by ELSIA kit and used to calculate cell viability. (**b**) The in vitro cytotoxicity of BiFab^CD20/CD3^ was analyzed by Annexin V/PI apoptosis detection kit, by using the same condition as described in (**a**). (**c**) Study on potential Fc-related cytotoxicity of BiFab^CD20/CD3^. The K562 cells and PBMCs were co-cultured with serial concentrations of non-binding IgG-based bispecific antibody or BiFab. The apoptosis rate was determined by Annexin V-Cy5 Apoptosis Detection Kit. (**d**) The in vitro cytotoxicity of BiFab^Her2/CD3^. Target tumor cells (SK-OV-3 or MDA-MB-468) and PBMC (E:T = 4:1) were incubated with serial concentrations of BiFab^Her2/CD3^ for 72 h, and the LDH release in the supernatant was determined by LDH detection kit. All data were shown as mean ± SD, *n* = 3. (**e**) The in vivo antitumor activities of BiFab^CD20/CD3^ in mouse xenograft model. Mice were inoculated subcutaneously with 2.5 × 10^6^ Ramos cells in the presence of 1 × 10^7^ fresh human PBMC from healthy donors at an E:T ratio of 4:1. All samples were administered intravenously via the tail vein at following dosages, 1 mg/kg of Fab^CD3^ and 1 mg/kg or 3 mg/kg of BiFab^CD20/CD3^ at every two days for four times.

**Figure 4 cancers-13-04540-f004:**
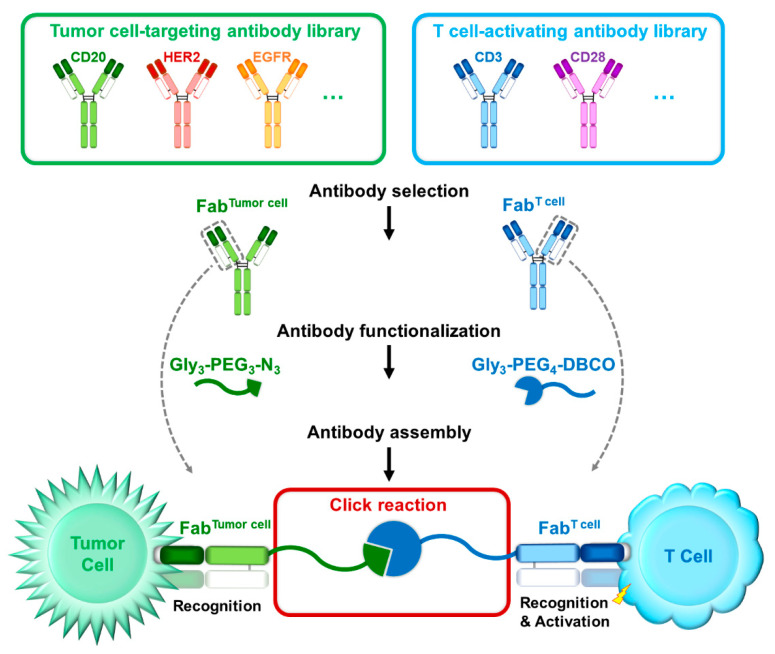
Schematic diagram of modular BiFab generation. Fabs could be adapted from full-length IgGs targeting tumor antigens or T cell/NK cell activating receptors. Fabs are genetically modified to have a C-terminal sortase A recognition motif (e.g., LPETG). Then, the paired click chemistry could be installed to the Fabs via sortase A mediated transpeptidation, followed by click reaction between two Fabs to generate BiFab.

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
