# Peer review of "Facile Generation of Potent Bispecific Fab via Sortase A and Click Chemistry for Cancer Immunotherapy"

_cancers, 2021, doi:10.3390/cancers13184540_

Round 1
Reviewer 1 Report
The manuscript (cancers-1361297) by Chen and Pan discovered the construction for bispecific Fab (BiFabCD20/CD3 and BiFabHer2/CD3) through chemo-enzymatic conjugation along with concrete experimental validations of their biological activities. The course of the study is performed systematically, and the description of the derived results are enough to support the potentiality of hybrid conjugates for application to antitumor treatment. The concise work is well organized and the data are consisted with the conclusion. Therefore, I recommend the acceptance of this manuscript after minor revision. The following are the details the author should consider before publication.
- When describing sortase A, what determines whether S is uppercase or lowercase?
- On page 3, line 89-91; expected that the vaguely expressed part (transfer the acyl, nucleophilic substrate) will be expressed more clearly.
- In identifying the structure of the adduct shown by the author in Figure 1.a; I wonder if the author used any auxiliary method (eg. NMR) other than checking the molecular weight to confirm the conformation of the structure of triazole generated through click chemistry between azide and alkyne.
- Should be consistent in citing and numbering references in the draft.
Author Response
Please see the attachment. And the "author-coverletter" attachment here is a wrong version of response, the right version of response is the attachment of "BiFab respond to reviewer 1"

Reviewer 2 Report
References page 14: References 28-38 are not included in the reference list. This makes reading and reviewing the discussion part of the manuscript very difficult.
Figure 3 page 11, should be labelled figure 4. It’s correctly labelled figure 4 in the text.
Discussion, line 356-357 “while spared CD20-negative tumor cells in vitro (Fig. 3a, 3b).” The results using the CD20 negative cell line is shown in Fig. 3c?
Materials and Methods
Line 110-111: “Anti-CD2 Fab sequence was derived from OKT3.” Can you please add a reference?
I cannot find a reference for the anti-HER2 Fab sequence, where is it derived from?
2.2. The section on the expression of the Fab fragments is very limited in detail. Please describe the expression in more detail, maybe in SI?
What does figure 1e show? Results page 6: line 238-239: “The purity of BiFab CD20/CD3 was further confirmed by RP-HPLC analysis (Figure 1e).” But in the caption for figure 1e (page 8, line 308-309) it says “Size exclusion chromatography (SEC) analysis of the purified BiFabCD20/CD3.”
Can you add examples of chromatograms from the SEC purification of the BiFabs to the Supplementary Information? It would give the reader a chance to evaluate the efficiency of the click-reaction. You write (page 6, lines 234-236) “Click reaction between FabCD3-DBCO and FabCD20-N3 at a molar ratio of 1:1 efficiently generated BiFabCD20/ CD3.”, but can you please show examples from the SEC purification? Or maybe an SDS-PAGE gel of the unpurified reaction mixture after the click-chemistry reaction?
In section 2.4 (page 4, line 154) you mention that you use “mhuOKT3” for your flow cytometry experiments. Is it the mAb OKT3 mentioned on line 111 on the same page? I don’t see any data for OKT3 in Figure 2a? It would be nice to see flow cytometry data for the corresponding antibodies, Ofatumumab and OKT3 respectively, so we can compare it to the binding abilities of the BiFab and the Fabs.
As you point out on page 6, lines 246-248 “Interestingly, BiFabCD20/CD3 had a higher binding affinity compared to that of FabCD20 or FabCD3 monomers to target cells (Figure 2a).”, it’s interesting to see that the BiFab seem to bind better to cells than the individual Fabs. Can you discuss if these results are expected or not?
Reviewer 3 Report
The manuscript cancers-1361297, Facile generation of potent bispecific Fab via Sortase A and Click chemistry for cancer immunotherapy, presents an interesting and valuable research. The review indicates some observations and suggestions for the authors to improve their work.
There are some editing correction needed to be done. For example, the way in which the authors are presenting the references. See for example rows 69, 72 or 109.
In the 2.2. section, the authors should detail on Sortase A. See for example: Targeting Bacterial Sortases in Search of Anti-Virulence Therapies with Low Risk of Resistance Development, Pharmaceuticals, 2021 Apr 30;14(5):415 and Sortase A: A Model for Transpeptidation and Its Biological Applications, Annu Rev Cell Dev Biol. 2018 Oct 6;34:163-188. The authors should also detail on what sortase A (probably that of S. aureus) they used and how it was prepared and purified. The reference mentioned as 28 is missing. Similarly on row 129, the references 28 and 29 are missing.
On row 137, check the section “reaction between azide (Fab-N3)”. Chemically it would be more exact to call it an azide derivative, and not just azide.
On row 145 and in some other places: use the subscript for the numbers in the chemical formulas. On row 153, and maybe in other sections, 6 should be superscript.
Row 146, check if it is correct 0% for isopropanol or the buffer.
The authors should try to keep a unitary style in all the manuscript. See for example “48 hours” on row 195 and “48 hrs” on row 200. See also “His” or “HIS”. Chose just one in the whole paper. The use or not of a space between a value and its unit of measure. In figure 1a, the authors use N3, but in figure 3, they use azide. Check the figures.
In section 3.1.1 the authors need to explain better the results. How were considered the optimal conditions? Based on the reactions yield? The authors should present the yields as % values based on the final product. Check figure 1. For example, in figure 1c on the horizontal scale should be minutes. In figure 1e the peek is not perfect analytically speaking. The authors should comment in the paper on this issue.
Round 2
Reviewer 2 Report
The authors have adequately answered the reviewer's criticisms.
The English language needs revision for minor details.
Author Response
We are very grateful for the comments and ideas presented by the reviewer. Please find below our responses to the reviewers´ queries, and our actions taken to revise the manuscript accordingly.
Comments to authors:
The authors have adequately answered the reviewer's criticisms.
Point 1: The English language needs revision for minor details.
Response 1: Thank you for remind us to check the manuscript thoroughly. By carefully checking the whole manuscript, we have revised some mistakes in numbers and writing formats. And we edited some sentences in the section of method, in order to make ourselves clearer. Please check the new version of our manuscript.